# Light-Independent Nitrogen Assimilation in Plant Leaves: Nitrate Incorporation into Glutamine, Glutamate, Aspartate, and Asparagine Traced by ^15^N

**DOI:** 10.3390/plants9101303

**Published:** 2020-10-02

**Authors:** Tadakatsu Yoneyama, Akira Suzuki

**Affiliations:** 1Department of Applied Biological Chemistry, Graduate School of Agricultural and Life Sciences, University of Tokyo, Yayoi 1-1-1, Bunkyo-ku, Tokyo 113-8657, Japan; 2Institut Jean-Pierre Bourgin, Institut national de recherche pour l’agriculture, l’alimentation et l’environnement (INRAE), UMR1318, RD10, F-78026 Versailles, France

**Keywords:** chloroplasts/plastids, dark nitrate assimilation, Fd-dependent glutamate synthase, leaves, nitrite-to-ammonia reduction, reductant-supplying system

## Abstract

Although the nitrate assimilation into amino acids in photosynthetic leaf tissues is active under the light, the studies during 1950s and 1970s in the dark nitrate assimilation provided fragmental and variable activities, and the mechanism of reductant supply to nitrate assimilation in darkness remained unclear. ^15^N tracing experiments unraveled the assimilatory mechanism of nitrogen from nitrate into amino acids in the light and in darkness by the reactions of nitrate and nitrite reductases, glutamine synthetase, glutamate synthase, aspartate aminotransferase, and asparagine synthetase. Nitrogen assimilation in illuminated leaves and non-photosynthetic roots occurs either in the redundant way or in the specific manner regarding the isoforms of nitrogen assimilatory enzymes in their cellular compartments. The electron supplying systems necessary to the enzymatic reactions share in part a similar electron donor system at the expense of carbohydrates in both leaves and roots, but also distinct reducing systems regarding the reactions of Fd-nitrite reductase and Fd-glutamate synthase in the photosynthetic and non-photosynthetic organs.

## 1. Introduction

Plants use inorganic nitrogen present in the soil for their growth mainly in the form of nitrate (NO_3_^−^). Following the absorption through the roots, the most oxidized form of NO_3_^−^ (+6) is reduced to organic forms (–2) such as amino acids prior to their incorporation into proteins, nucleotides, and chlorophylls. Although plants can assimilate NO_3_^−^ to amino acids in both the photosynthetic leaves and non-photosynthetic roots, the major sites of nitrate assimilation are green shoots where energy (ATP), reductant (electrons) and organic skeletons are produced by photosynthetic reactions using solar energy. In addition, the nitrate assimilation in the leaves takes place in the night, i.e., by using storage carbohydrates [1]. However, the activities of nitrate assimilation in darkness measured from the 1950s to 1970s were fragmental and variable. Here, we present physiological aspects of light-independent nitrate-to-asparagine assimilation in the leaves by referring to the light-independent nitrate assimilation in the non-photosynthetic roots at the expense of carbohydrates, supplied by transport from the shoots, as reviewed recently [2].

## 2. ^15^N Tracing Analysis of Dark Nitrate Assimilation into Amino Acids in Leaves

Direct evidence of dark nitrate assimilation into amino acids in the leaves was obtained by incubating the leaf tissues with ^15^N-nitrate in darkness. Delwiche [3] fed detached immature leaves of tobacco (*Nicotiana tabacum*) with ^15^N-KNO_3_ solution (15 atom % excess ^15^N) for 24 h via petioles in darkness, and detected ^15^N-labelling in ammonia plus amide-N, and in ammonia fraction by 2.89 and 0.61 atom % excess ^15^N, respectively. Mendel and Visser [4] conducted a short incubation (30 min) of tomato leaf discs with ^15^N-KNO_3_ (14 atom % excess ^15^N) in the dark, and detected ^15^N-labelling in free ammonia fraction in duplicated samples by 2.13 and 2.42 atom % excess ^15^N. Canvin and Atkins [5] incubated leaf segments of wheat (*Triticum aestivum* L.) and corn (*Zea mays* L.) with ^15^N-NaNO_3_ solution (95 atom % ^15^N) for 15 and 30 min in darkness. After 15 min, they detected little ^15^N enrichment in the soluble amino acid fraction in darkness compared with the high ^15^N labelled amino acids in the light and concluded that the nitrate assimilation was strictly dependent on the light.

Shortly after the findings of ^15^N-labelled amino acids in the roots of rice (*Oryza sativa* L.) seedlings by feeding with ^15^N-ammonium [6] and ^15^N-nitrate [7], Ito and Kumazawa [8] conducted light and dark incubation of leaf discs of sunflower (*Helianthus annuus* L.) with ^15^NO_3_^−^, ^15^NO_2_^−^, and ^15^NH_4_^+^ for 30 min. They detected ^15^N-labelled glutamine (amino-N and amide-N), glutamate, and aspartate as shown in Table 1. In these studies, ^15^N-labelling of individual amino acids was determined by a combination of their separation on two-dimension thin-layer chromatography and ^15^N enrichment by emission optical spectrometry, as first described by Yoneyama and Kumazawa [6]. Dark ^15^N-labelling in the amino-N of glutamine, glutamate, and aspartate was less than that in the light, while the dark ^15^N-labelling of the amide-N of glutamine was higher than that in the light irrespective of the feeding with ^15^NO_3_^−^, ^15^NO_2_^−^, or ^15^NH_4_^+^. The results indicated that the transfer rate of the glutamine amide-N to 2-OG forming glutamate in darkness was lower than in the light although nitrate was reduced sequentially to nitrite and ammonia in darkness. It is noteworthy that in the light, the ^15^N-labellings of glutamine amide-N from all of ^15^NO_3_^−^, ^15^NO_2_^−^, and ^15^NH_4_^+^ were apparently lower comparing with those in darkness. This can be explained by a pool of NH_4_^+^ diluted by a large amount of photorespiratory NH_4_^+^ [9] and assimilated into glutamine amide-N by glutamine synthetase (Table 1). The non-photosynthetic root tissues, which have no photorespiratory activity, actively assimilated ^15^N into ^15^N-amide of glutamine and glutamate by the root feeding with ^15^NO_3_^−^ or ^15^NO_2_^−^ [2] as observed in the leaves in darkness (Table 1).

In the 1970s, a simple in vivo assay of nitrate reductase activity was widely employed using leaf segments without extraction of enzymes. In this technique, it was assumed that nitrite, nitrate reductase reaction product, was barely assimilated when the assay was carried out in the dark [10,11,12]. In vivo nitrate reductase assays were conducted using ^15^N-NaNO_3_ to measure its reduction to nitrite and the assimilation into amino acids under aerobic or anaerobic conditions [13]. Nitrite production was active under anaerobic conditions while ^15^N incorporation into amino acids was intensive under aerobic conditions than under anaerobic conditions (Table 2).

The ^15^N-labelling experiments were carried out using green and white chlorophyll-less leaves of albino mutant seedlings of rice, which were produced by chemical mutation. It was shown that the white leaves fed with ^15^NO_2_^−^ in darkness [14] had lower ^15^N-enrichments (atom % excess ^15^N) in glutamine, glutamate, and aspartate than green leaves, but a large ^15^N accumulation occurred in glutamine and particularly in asparagine in the white leaves (Table 3). In vitro activities of both nitrate reductase and nitrite reductase were detected in the leaf extracts of normal and chlorophyll-less leaves of albino mutant seedlings of barley (*Hordeum vulgare* L.), although their specific activities were less in the chlorophyll-less leaves than in the normal leaves [15].

## 3. Enzymes for Nitrate Assimilation for Nitrate Assimilation in Leaves

### 3.1. Nitrate Reduction to Ammonia

Nitrate is not only an essential nutrient but also a signaling molecule of cellular events in response to its fluctuating availability in both space and time. Nitrate is taken up into the roots by the nitrate transporters on the plasma membrane and regulates lateral root developments [16]. Transported in the xylem, nitrate is distributed within the plant by the nitrate transporters located in the shoots, leaves, flowers and seeds, and triggers expression of nitrate-responding genes, leaf development, and seed germination [17,18,19]. Nitrate reductase in the cytosol (Equation (1)) and ferredoxin (Fd)-dependent nitrite reductase in the chloroplasts/plastids (Equation (2)) catalyze the sequential reactions of nitrate reduction to nitrite and nitrite to ammonia, respectively (Figure 1).

Nitrate reductase (NADH-NR, EC 1.6.6.1; NAD(P)H-NR, EC 1.6.6.2)

                     2e
NO_3_^−^ + NADH or NAD(P)H + H^+^ → NO_2_^−^ + NAD^+^ or NADP^+^ + H_2_O(1)

Nitrite reductase (NiR, EC 1.7.7.1)

                      6e
NO_2_^−^ + 6 Fd_red_ + 8 H^+^ → NH_4_^+^ + 6 Fd_ox_ + 2 H_2_O(2)

NADH produced by glycolysis served as a major electron donor to nitrate reductase (NADH-NR, EC 1.6.6.1) in the leaves of most plant species [20,21], while a bi-specific nitrate reductase to NADH and NADPH (NAD(P)H-NR, EC 1.6.6.2) was found in the leaves of soybean (*Glycine max*. Merr.) [22,23] and barley [24]. NR, located in the cytosol as homodimer or homotetramer [25], utilizes two electrons from NAD(P)H to reduce nitrate. The enzyme has three functional domains for binding of the prosthetic group or cofactor: flavin adenine dinucleotide (FAD), heme-Fe, and molybdate-pterin (MoCo) [17,26].

Ferredoxin-dependent NiR (Fd-NiR), localized in the chloroplasts/plastids [25,27,28,29], utilizes six electrons from photo-reduced ferredoxin (Fd) as the electron donor to reduce nitrite to ammonia. Fd-NiR has two prosthetic groups: a siroheme and an iron-sulfur cluster, linked by one of four cysteine residues of the iron-sulfur cluster [26,30]. The high NiR activity, assayed by the disappearance of nitrite or formation of ammonia, was detected in the presence of strong reducing dyes methyl viologen or benzyl viologen reduced chemically by dithionite [31,32]. NiR activity in vitro was found in the presence of Fd reduced by ferredoxin: NADP^+^ oxidoreductase (FNR, EC 1.18.1.2) depending on NADPH, which was generated by a diaphorase containing glucose 6-phosphate dehydrogenase (G6PDH, EC 1.1.1.49) [33]. The in vitro titration analysis showed that the NiR may make a complex close to 1:1 with reduced Fd for the efficient electron transfer [34].

### 3.2. Glutamine Synthesis and Metabolism to Glutamate and Asparagine in Leaves

Up to 1974, it was generally accepted that ammonia assimilation is catalyzed by ammonia-inducible glutamate dehydrogenase (GDH, EC 1.4.1.2), which catalyzes a reversible amination of 2-oxoglutarate by ammonia generating L-glutamate and its conversion to ammonia and 2-oxoglutarate. Under conditions of ammonia excess, ammonia was assimilated into glutamine by glutamine synthetase (GS or L-glutamate:ammonia ligase (ADP), EC 6.3.1.2, Equation (3)), and under more excessive levels of ammonia and glutamine, asparagine served as an storage compound of nitrogen via the catalysis by glutamine-dependent asparagine synthetase (AS, EC 6.3.5.4).

However, Tempest et al. [35] reported a new pathway of ammonium assimilation by the coupled reactions of GS and NADPH-dependent glutamate synthase (L-glutamine (amide): 2-oxoglutarate aminotransferase: NADPH-GOGAT, EC 1.4.1.13) in bacteria. In plants, glutamate synthase activity with ferredoxin (Fd-GOGAT, EC 1.4.7.1) (Equation (4)) in the chloroplasts was reported by Lea and Miflin [36]. Extensive studies provided evidence for the operation of GS (GS2)/GOGAT (Fd-GOGAT) cycle in the chloroplasts/plastids as the major route of primary nitrogen assimilation [37] and photorespiratory ammonium re-assimilation in leaves [9,38,39,40,41,42].

Glutamine synthetase (GS, EC 6.3.1.2)
L-Glutamate + NH_3_ + ATP → L-Glutamine + ADP + Pi(3)

Fd-glutamate synthase (Fd-GOGAT, EC 1.4.7.1)

                          2e
L-Glutamine + 2-Oxoglutarate + Fd_red_ → 2 L-Glutamate + Fd_ox_(4)

^15^N tracing studies confirmed an in vivo operation of GS2/Fd-GOGAT cycle for the nitrogen assimilation from nitrate, nitrite, and ammonia into amino acids in the light or in darkness (see Table 1).

The GS occurs in two forms, cytosolic GS1 and plastidial GS2, in both leaves and roots with different ratio according to plants [43]. The cytosolic GS1 in the senescent leaves may function to assimilate a high level of ammonia during nitrogen remobilization [44,45]. Two forms of GOGAT, Fd-GOGAT and NADH-GOGAT (EC 1.4.1.14), are distinguished in leaves and roots of different plant species. The Fd-GOGAT in vitro activity was found active in the light-grown mature leaves, and the isolated chloroplasts showed a high activity of Fd-GOGAT in the light and low activity in darkness [46,47]. The enhancement of Fd-GOGAT activity and the Fd-GOGAT protein level during the greening of the etiolated plants [48] via a reversible red/far-red reaction provided evidence for a regulation mediated by the phytochromes [39,49]. Two molecules of glutamate are formed from glutamine and 2-OG through the intramolecular reactions of NH_2_-releasing glutaminase and 2-OG transamidation with -NH_2_ using reduced Fd (Equation (4)) [50,51,52].

In the cytosol of leaf cells, the amide-N of glutamine is transferred to aspartic acid to form asparagine by asparagine synthetase utilizing ATP (AS, Equation (5)). AS could use both ammonia and glutamine-amide while glutamine is a preferred amide donor. Km for glutamine (0.04–1.0 mM) was 40-fold lower than for ammonium ion [53]. The accumulation of asparagine [54] and AS-mRNA of Class I *ASN* genes [55,56,57] was enhanced in darkness. This is consistent with a carbon supply by an anaplerotic reaction of cytosolic phosphoenolpyruvate carboxylase [58,59]. Oxaloacetate thus formed is transaminated with glutamate by aspartate aminotransferase to aspartate, substrate of AS (AspAT, Equation (6)).

Asparagine synthetase (AS, EC 6.3.5.4)
L-Glutamine + L-Aspartate + ATP → L-Asparagine + L-Glutamate + AMP + PPi(5)

Aspartate aminotransferase (AspAT, EC 2.6.1.1)
L-Glutamate + Oxaloacetic acid ⟷ L-Aspartate + 2-Oxoglutarate(6)

AspAT in plants exists as isoforms, which are located in different subcellular compartments. The *ASP2* mRNA for cytosolic AspAT2 in Arabidopsis was most abundantly expressed in root tissue and accumulated at higher levels in illuminated leaves and dark-adapted leaves [60], indicating that AspAT2 may be involved in synthesizing aspartate pool for asparagine synthesis by AS2 in dark-adapted plants.

Figure 1 depicts the scheme of nitrate assimilation in darkness in the leaf cells. Nitrate delivered from the xylem is reduced to NO_2_^−^ by the cytosolic NR using NADH from glycolysis. Nitrite is diffused into the chloroplasts [61] and reduced to ammonia by Fd-NiR. The ammonium is assimilated to glutamine by chloroplast-localized GS2 using ATP imported [62] and then to glutamate by Fd-GOGAT using 2-OG produced via cytosolic NADP-dependent isocitrate dehydrogenase (NADPH-ICDH, EC 1.1.1.42) [63,64,65] and/or in part by mitochondrial NAD-dependent isocitrate dehydrogenase (NADH-IDH, EC 1.1.1.41) [66]. Glutamate and glutamine in the chloroplasts are released to the cytosol and glutamate is metabolized to aspartate by cytosolic AspAT2. Aspartate thus produced is combined with the amide of glutamine, forming asparagine by leaf cytosolic AS2. The nitrogen assimilation pathway from nitrate to asparagine catalyzed by Fd-NiR, GS, Fd-GOGAT, AspAT and AS in darkness in the green leaves was in line with ^15^NO_2_^−^ tracing data shown in Table 3. In chlorophyll-less white leaves, which contained proplastids [14], the activity of AS2 (asparagine formation) might be higher than the Fd-GOGAT activity (glutamate formation), suggesting a low level of Fd-GOGAT without light [48].

## 4. Reductant-Supplying Systems for Dark Nitrogen Assimilation in Leaves

### 4.1. Reductant Supply for Nitrate Reductase and Nitrite Reductase

NADH-NR (Equation (1)) from spinach (*Spinacia oleracea*) had a midpoint redox potentials (*E*m_7_) of −60 mV and *E*m_0_ of NADH being −320 mV [26]. NADH can be generated in the cytosol by the respiratory network through glyceraldehyde-3-phosphate dehydrogenase (GAPDH, EC 1.2.1.12) or through an anaplerotic reaction by malate dehydrogenase (NAD-MDH, EC 1.1.1.37) [21].

Following nitrite import from the cytosol into the chloroplasts/plastids by passive diffusion, nitrite is reduced to ammonia by NiR. This reaction involves a flow of six electrons from Fd_red_ → ((4Fe‒4S) → siroheme) → NO_2_^−^ (Equation (2)). Ferredoxins are reduced by photosystems and discovered in 1963 in the leaves of *Cucurbita pepo* [27] and spinach leaves [28,29]. The *E*m of Fd siroheme for NO_2_^−^-binding and that of (4Fe-4S) cluster of spinach NiR were determined to be −290 mV and −365 mV, respectively [67]. In maize, four Fd iso-proteins were identified showing the tissue-preferential distribution: leaf-specific and light-inducible FdI (*E*m = −423 mV) and FdII (*E*m = −406 mV) in leaves; FdIII (*E*m = −345 mV) and FdIV in all plant parts including roots [68,69,70,71,72] (Table 4).

Under the light, leaf NiR received electrons from Fd reduced in the photosystem I (PSI), while in darkness without the light energy, stromal Fd received electrons via FNR from the primary electron donor NADPH (*E*m = −320 mV, [76]), as shown in Figure 2. In darkness, plastidial NADPH was generated by oxidative pentose phosphate pathway using G6PDH (EC 1.1.1.49) and 6-phosphogluconate dehydrogenase (6PGDH, EC 1.1.1.44) at the expense of glucose 6-phoshate (G6P) produced from starch in the plastids and G6P imported from the cytosol to the plastids [77,78,79,80,81]. Thus, FNR (EC 1.18.1.2, *E*m = ~ −320 mV, [82]) in leaves and roots could catalyze the reversible electron transfer between NADPH and leaf-type Fd (e.g., FdI in maize) and root-type Fd (e.g., FdIII in maize) (see Table 4) as shown in Equation (7) [83,84]. In the leaves, both leaf-type and root-type FNRs were found in the stroma of the leaf chloroplasts as well as some leaf-type FNRs in the thylakoid membrane [73,74,75], and gene expression for the root-type FNRs was induced by nitrate feeding [18,73]. In vitro electron donation from NADPH to maize FdIII:R-FNR complex was active with a ratio of 1.00, in contrast to a lower ratio (0.68) in the leaf NADPH‒FdI:L-FNR system [85].
2Fd (Fe^3+^) + NADPH ⟷ 2Fd (Fe^2+^) + NADP^+^ + H^+^(7)

The previous investigations of ^15^NO_3_^−^ and ^15^NO_2_^−^ reduction and assimilation into amino acids in leaf segments in darkness showed high magnitudes than in the light in tobacco [3], tomato [4], sunflower [8], and rice [14], while negligible in wheat and maize leaves [5]. Such difference in magnitudes may be derived from an availability of electron in NADPH-FNR-Fd systems under changing electron donating system in darkness (Figure 2).

Under anaerobic condition, the nitrite accumulation was large and glutamine quantity became small (Table 2). These results were caused by the elevated level of NO_3_^−^-reducing NADH and the shortage of NADPH [12,86]: The NADPH deficiency may reduce nitrite-to-ammonia reduction and induce the glutaminase activity of GOGAT, causing disappearance of glutamine [85]. Expression studies determined the level of Fd and FNR in the plants deprived of light by continuous or prolonged darkness. This dark stress declined the photosynthetic FNR subforms (*L*FNRI and *L*FNRII) at both mRNA and protein levels at the base section of wheat leaf in the presence of nitrate [87], and leaf-type Fds (Fd I and Fd II) in maize leaves [68], suggesting a less efficient contribution of NADPH-FNR-Fd system necessary to the reactions of Fd-NiR and Fd-GOGAT. Exposure of Arabidopsis to environmental stress such as extended high light (120 h at approximately 500 micromole photons m^2^ sec^1^) resulted in a gradual decrease of AtFd2 (At1g60950) in its mRNA (to 10% of the WT level) and protein (to 13%) [88]. Down-regulation or mutation of Fd in Arabidopsis [89] and potato (*Solanum tuberosum*) [90] caused an inactivated photosynthesis and inhibited plant growth.

### 4.2. Reductant Supply to the Fd-Dependent GOGAT

Ammonia produced by NiR in the plastids is assimilated into glutamine by GS2 using energy (ATP) from mitochondria [62,79]. The glutamine amide-N is transferred to 2-OG, yielding two molecules of glutamate takes place in the chloroplasts/plastids by Fd-GOGAT [36,52,91]. Fd-GOGAT is a flavin and iron-sulfur-containing protein. The isopotential of these chromophore and cluster were reported to have *E*m of −225 ± 10 mV in the enzyme from spinach leaves [92]. NADPH, generated by the oxidative pentose phosphate pathway, was also shown to be a primary electron donor for the reactions of Fd-GOGAT in darkness [80].

A ^15^N-tracing study in sunflower leaf discs (Table 1) showed that the activity of glutamine amide-N transfer to 2-oxoglutamate forming glutamate in darkness was less active than in the light, where PSI supplied electrons to Fd (see Figure 2). The activity of Fd-GOGAT in vegetable leaves was the major regulating step of nitrate assimilation in the whole plant [1].

## 5. Conclusions

Leaves represent a major site of primary nitrogen assimilation in concert with roots, photorespiratory NH_4_^+^ re-fixation, and translocation of nitrogen within the plant. ^15^N-tracing studies with leaves demonstrated that nitrate was reduced to ammonia and assimilated into glutamine, glutamate, aspartate, and asparagine in the light and in darkness. In the present review, we examined that the reductive incorporation of nitrate into amino acids occurs in darkness in the leaves through the isoforms of NR, NiR, GS, Fd-GOGAT, AspAT, and AS. To provide reducing equivalents to the NiR and Fd-GOGAT reactions in the dark, single leaf contains the photosynthetic form of FNR and Fd and heterotrophic form of FNR and Fd, indicating inter-connected electron supply systems in the light and in darkness. It remains to dissect the operation mechanism of electron donation systems in distinct types of photosynthetic cells and heterotrophic cells of a leaf.

## Figures and Tables

**Figure 1 plants-09-01303-f001:**
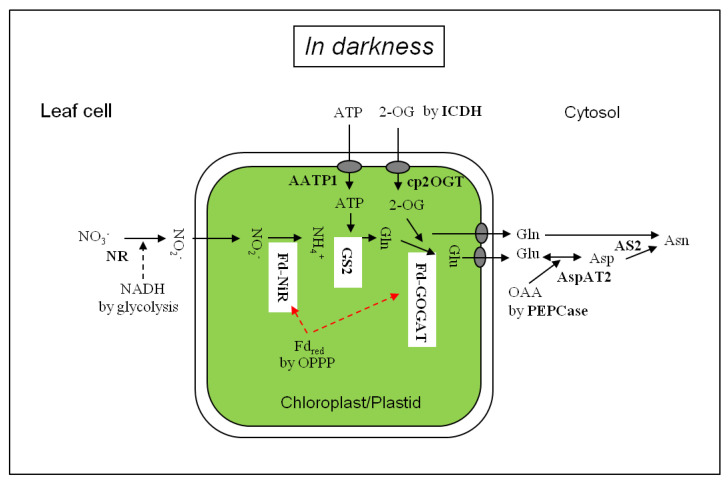
Nitrate and nitrite assimilation in the leaf cells in darkness. AATP1, ATP/ADP transporter; AS, asparagine synthetase; AspAT: aspartate aminotransferase; cp2OGT, chloroplastic 2-OG transporter; Fd_red_, reduced ferredoxin; Fd-GOGAT, Ferredoxin-dependent glutamate synthase; Fd-NiR, Ferredoxin-dependent nitrite reductase, GS2: chloroplastic glutamine synthetase; ICDH, NADPH-isocitrate dehydrogenase; NR: nitrate reductase; PEPCase, phosphoenolpyruvate carboxylase; OAA, oxaloacetate; OPPP, oxidative pentose phosphate pathway.

**Figure 2 plants-09-01303-f002:**
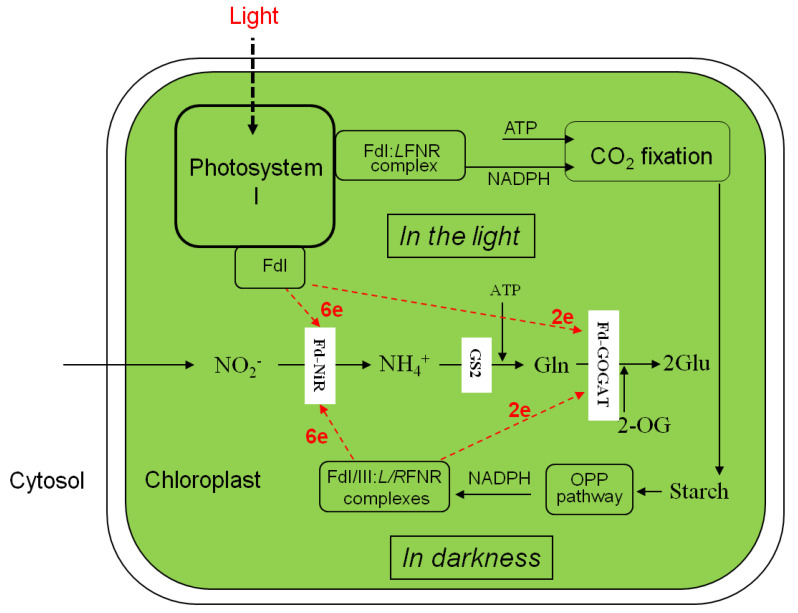
Nitrite assimilation and glutamate formation in the chloroplasts in the light and darkness. *L*FNR, leaf-type FNR; *R*FNR, root-type FNR. The other abbreviations in Figure 1 legend.

**Table 1 plants-09-01303-t001:** The ^15^N atom % excess of amino acids extracted from the sunflower leaf discus treated with 1 mM ^15^NO_3_^−^, ^15^NO_2_^−^, and ^15^NH_4_^+^ for 30 min under darkness as compared to those under the light (20,000 lux).

Amino Acids	^15^NO_3_^−^		^15^NO_2_^−^		^15^NH_4_^+^	
	Dark	Light	Dark	Light	Dark	Light
Glutamine						
Amino-N	0.15	0.35	1.4	1.82	8.6	14.8
Amide-N	1.53	0.88	17.5	5.80	59.4	35.9
Glutamate	0.13	0.40	2.3	4.05	8.5	15.2
Aspartate	0.15	0.36	2.0	4.03	8.3	16.4

^15^N-labeling (atom % excess) of the substrates were 98.6 for K^15^NO_3_, 98.8 for Na^15^NO_2_, and 97.1 for (^15^NH_4_)_2_SO_4_. Adapted from Ito and Kumazawa [8].

**Table 2 plants-09-01303-t002:** Nitrite formation and ^15^N incorporation into amino acids in the wheat leaf segments treated with 50 mM Na^15^NO_3_^−^ for 60 min under aerobic (in air) or anaerobic (in N_2_) conditions in darkness.

Atmosphere	Aerobic	Anaerobic
Nitrite formation (μg N g^−1^ fr wt)	2.1 ± 0.7	46 ± 2
^15^N incorporation (atom % excess ^15^N) in		
Glutamine	3.74	Small quantity
Glutamate	0.92	0.22
Asparagine	0.70	0.01
Aspartate	0.88	0.06

^15^N-labeling (atom % excess) of Na^15^NO_3_ was 100. Adapted from Yoneyama [13].

**Table 3 plants-09-01303-t003:** Assimilation of 1 mM ^15^NO_2_^−^ by 60-min incubation to amino acids in the green and white leaf segments from 15-day-old albino mutant rice plants in darkness.

Amino Acid	Amino Acid Content (μ mol g^−1^fr wt)		^15^N-Enrichment (Atom % Excess)		^15^N-Content (μg N g^−1^fr wt)	
	Green	White	Green	White	Green	white
Glutamine	3.7	8.6	7.30	2.25	7.56 (60%)	5.42 (68%)
Glutamate	7.1	4.1	3.05	0.55	3.03 (24%)	0.32 (4%)
Asparagine	3.4	47.1	0.92	0.16	0.87 (7%)	2.11 (26%)
Aspartate	4.1	4.1	2.14	0.26	1.23 (10%)	0.15 (2%)

^15^N-labeling (atom % excess) of Na^15^NO_2_ was 99.2. The numerals in the parenthesis are % distribution of ^15^N in four amino acids. Adapted from Yoneyama [14].

**Table 4 plants-09-01303-t004:** Representative iso-proteins of Fd and FNR identified in the leaves and roots of maize and Arabidopsis.

Plant Species	Fd Species	FNR Species	References
Maize			[68,71,73,74]
Leaves	FdI, FdII, FdIII, FdIV, FdVI	*L*FNR1, *L*FNR2, *L*FNR3, *R*FNR	
Roots	FdIII, FdIV, FdVI	*R*FNR	
Arabidopsis			[75]
Leaves	Fd1, Fd2, Fd3	*L*FNR1, *L*FNR2, *R*FNR1, *R*FNR2	
Roots	Fd3	*R*FNR1, *R*FNR2	

*L*, leaf-type; *R*, root-type.

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
