# Peer review of "Light-Independent Nitrogen Assimilation in Plant Leaves: Nitrate Incorporation into Glutamine, Glutamate, Aspartate, and Asparagine Traced by 15N"

_plants, 2020, doi:10.3390/plants9101303_

Round 1

Reviewer 1 Report

Comments and Suggestions for Authors

The manuscript titled “Reflection of Studies over the Past Years on the Light-Independent Nitrogen Assimilation in Plant Leaves: Nitrate Incorporation into Gln, Glu, Asp and Asn Traced by 15N” Manuscript ID: plants-937384, by Yoneyama and Suzuki, submitted for publication to Plants, is an interesting review article on the physiological mechanism(s) of light-independent nitrate-to-asparagine assimilation in the leaves at the expense of carbohydrates.

Major concerns

  • Although the work is well-planned, however, the manuscript is relatively high plagiarized in some parts. I detected some plagiarism (about 29%) throughout the manuscript. Some sections should paraphrase prior to the acceptance of the manuscript for publishing, even if they were copied from your previous work (Yoneyama and Suzuki, Plant Physiol. Biochem. 2019, 136, 245-254), particularly the paragraph lines 31-34, 116-117, 130-133, 142-148, 156-158, 163-168, and lines 223-232.
  • Another concern is about English use. The manuscript should be thoroughly revised and improved in terms of comprehensibility and appropriateness of scientific terms. For example, As a plant physiologist, I prefer to avoid the use of "higher" and "lower" plants; please use more precise terms such as "land plants", "angiosperms", "vascular plants", "bryophytes", etc. This issue should be revised throughout the manuscript.
  • Moreover, the manuscript should be carefully and deeply revised for grammar and English use, since many minor mistakes are found throughout the whole paper.

Other comments and suggestions

Title

  • I suggest shortening the title by deleting “Reflection of Studies over the Past Years on”. Moreover, please spell out all the abbreviations in the title (Gln, Glu, Asp, and Asn).

Abstract

  • I could recommend rewriting the abstract to be more informative rather than historical description.
  • Line 20: It is uncommon to cite references in the abstract, please remove it.
  • Line 22: unitalicize the word “via”.

Keywords:

  • Line 27-28: Capitalize the first letter of each keyword.

Introduction

Some other grammar mistakes were found throughout the manuscript (I’m not going to list all of them), please carefully revised the manuscript for grammar and English use.

Author Response

Reviewer 1

Comments and Suggestions for Authors

The manuscript titled “Reflection of Studies over the Past Years on the Light-Independent Nitrogen Assimilation in Plant Leaves: Nitrate Incorporation into Gln, Glu, Asp and Asn Traced by 15N” Manuscript ID: plants-937384, by Yoneyama and Suzuki, submitted for publication to Plants, is an interesting review article on the physiological mechanism(s) of light-independent nitrate-to-asparagine assimilation in the leaves at the expense of carbohydrates.

Major concerns

Although the work is well-planned, however, the manuscript is relatively high plagiarized in some parts. I detected some plagiarism (about 29%) throughout the manuscript. Some sections should paraphrase prior to the acceptance of the manuscript for publishing, even if they were copied from your previous work (Yoneyama and Suzuki, Plant Physiol. Biochem. 2019, 136, 245-254), particularly the paragraph lines 31-34, 116-117, 130-133, 142-148, 156-158, 163-168, and lines 223-232.

Response: The overlapping paragraphs are corrected as follows by referring to those underlined by the reviewer:

(a) lines 31-34:

Plants use inorganic nitrogen present in the soil for their growth mainly in the form of nitrate (NO3-). Following the absorption through the roots, the most oxidized form of NO3- (+6) is reduced to organic forms (-2) such as amino acids prior to their incorporation into proteins, nucleotides and chlorophylls.

(b) lines 116-117:

Nitrate reductase in the cytosol (Eq. (1)) and ferredoxin (Fd)-dependent nitrite reductase in the chloroplasts/plastids (Eq. (2)) catalyze the sequential reactions of nitrate reduction to nitrite and nitrite to ammonia.

(c) lines 130-133:

NR, located in the cytosol as homodimer or homotetramer [25], utilizes two electrons from NAD(P)H to reduce nitrate the enzyme has three functional domains for binding of the flowing prosthetic group or cofactor: flavin adenine dinucleotide [FAD], heme-Fe, and molybdate-pterin (MoCo) [17, 26].

(d) lines 142-148:

Ferredoxin-dependent NiR (Fd-NiR), localized in the chloroplasts/plastids [25, 27, 28, 29], utilizes six electrons from photo-reduced ferredoxin (Fd) as the electron donor to reduce nitrite to ammonia. Fd-NiR has two prosthetic groups: a siroheme and an iron-sulfur cluster, linked by one of four cysteine residues of the iron-sulfur cluster [26, 30]. The high NiR activity, assayed by the disappearance of nitrite or formation of ammonia, was detected in the presence of strong reducing dyes methyl viologen or benzyl viologen reduced chemically by dithionite [31, 32].

(e) lines 156-158:

Up to 1974, it was generally accepted that ammonia assimilation is catalyzed by ammonia-inducible glutamate dehydrogenase (GDH, EC 1.4.1.2), which catalyzes a reversible amination of 2-oxoglutarate by ammonia generating L-glutamate and its conversion to ammonia and 2-oxoglutarate.

(f) lines 163-168:

However, Tempest et al. [35] reported a new pathway of ammonium assimilation by the coupled reactions of GS and NADPH-dependent glutamate synthase [L-glutamine (amide) : 2-oxoglutarate aminotransferase: NADPH-GOGAT, EC 1.4.1.13] in bacteria. In plants, glutamate synthase activity with ferredoxin (Fd-GOGAT, EC 1.4.7.1) (Eq. (4)) in the chloroplasts was reported by Lea and Miflin [33]. Extensive studies provided evidence for the operation of GS (GS2)/GOGAT (Fd-GOGAT) cycle in the chloroplasts/plastids as the major route of primary nitrogen assimilation [37] and photorespiratory ammonium re-assimilation in leaves [9, 38, 39, 40, 41, 42].

(g) lines 223-232:

NADH-NR (Eq. (1)) from spinach (Spinacia oleracea) had a midpoint redox potentials (Em7) of ‒60 mV and Em0 of NADH being ‒320 mV [26]. NADH can be generated in the cytosol by the respiratory network through glyceraldehyde-3-phosphate dehydrogenase (GAPDH, EC 1.2.1.12) or through an anaplerotic reaction by malate dehydrogenase (NAD-MDH, EC 1.1.1.37) [21].

Following nitrite import from the cytosol into the chloroplasts/plastids by passive diffusion, nitrite is reduced to ammonia by NiR. This reaction involves a flow of six electrons from Fdred → [(4Fe‒4S) → siroheme] → NO2- (Eq. (2)). Ferredoxins are reduced by photosystems and discovered in 1963 in the photosynthetic Cucurbita pepo [27] and spinach leaves [28, 29]. The Em of Fd siroheme for NO2--binding and that of [4Fe-4S] cluster of spinach NiR were determined to be ‒290 mV and ‒365 mV, respectively [67].

Another concern is about English use. The manuscript should be thoroughly revised and improved in terms of comprehensibility and appropriateness of scientific terms. For example, As a plant physiologist, I prefer to avoid the use of "higher" and "lower" plants; please use more precise terms such as "land plants", "angiosperms", "vascular plants", "bryophytes", etc. This issue should be revised throughout the manuscript.

Response: “plants” is used throughout instead of “higher plants”:

(1) Introduction: 3rd paragraph, and

(2) “3.2 Glutamine synthesis and metabolism to glutamate and asparagine in leaves” - Aspartate aminotransferase, 3rd line.

Moreover, the manuscript should be carefully and deeply revised for grammar and English use, since many minor mistakes are found throughout the whole paper.

Response: English use and grammar are revised. It was also subjected to English correction service of MDPI.

Other comments and suggestions

Title

I suggest shortening the title by deleting “Reflection of Studies over the Past Years on”. Moreover, please spell out all the abbreviations in the title (Gln, Glu, Asp, and Asn).

Response: The title is changed to “Light-Independent Nitrogen Assimilation in Plant Leaves: Nitrate Incorporation into Glutamine, Glutamate, Aspartate and Asparagine Traced by 15N”.

Abstract

I could recommend rewriting the abstract to be more informative rather than historical description.

Response: The abstract is modified as follow:

Abstract: Although the nitrate assimilation into amino acids in photosynthetic leaf tissues is active under the light, the studies during 1950s and 1970s in the dark nitrate assimilation provided fragmental and variable activities, and the mechanism of reductant supply to nitrate assimilation in darkness remained unclear. 15N tracing experiments unraveled the assimilatory mechanism of nitrogen from nitrate into amino acids in the light and in darkness by the reactions of nitrate and nitrite reductases, glutamine synthetase, glutamate synthase, aspartate aminotransferase and asparagine synthetase. Nitrogen assimilation in illuminated leaves and non-photosynthetic roots occurs either in the redundant way or in the specific manner regarding the isoforms of nitrogen assimilatory enzymes in their cellular compartments. The electron supplying systems necessary to the enzymatic reactions share in part a similar electron donor system at the expense of carbohydrates in both leaves and roots, but also distinct reducing systems regarding the reactions of Fd-nitrite reductase and Fd-glutamate synthase in the photosynthetic and non-photosynthetic organs.

Line 20: It is uncommon to cite references in the abstract, please remove it.

Response: References in the abstract are deleted.

Line 22: unitalicize the word “via”.

Response: “via” is un-italicized.

Keywords:

Line 27-28: Capitalize the first letter of each keyword.

Response: The first letter of each key word is capitalized.

Introduction

Some other grammar mistakes were found throughout the manuscript (I’m not going to list all of them), please carefully revised the manuscript for grammar and English use

Response: English use and grammar are revised. They will be also checked by English correction service of MDPI.

Reviewer 2 Report

The review written by Yoneyama and Suzuki reports information in relation to the nitrogen assimilation under dark condition. The language used in the review is easy to understand. It recalls my basic concepts of Nitrogen assimilation. I will recommend the acceptance of this manuscript with some minor revisions.

1. The review lacks conclusion with further possibilities of research and questions to be answered in the the field of Nitrogen assimilation.

2. Just a suggestion, authors can include a section representing effect of abiotic stress on Nitrogen assimilation in light independent conditions or, Do stress hampered photosynthesis facilitates light independent N-assimilation?

Author Response

Reviewer 2

The review written by Yoneyama and Suzuki reports information in relation to the nitrogen assimilation under dark condition. The language used in the review is easy to understand. It recalls my basic concepts of Nitrogen assimilation. I will recommend the acceptance of this manuscript with some minor revisions.

  1. The review lacks conclusion with further possibilities of research and questions to be answered in the the field of Nitrogen assimilation.

[Response]

Response: Conclusion is included as follow:

“Leaves represent a major site of primary nitrogen assimilation in concert with roots, photorespiratory NH4+ re-fixation and translocation of nitrogen within the plant. 15N-tracing studies with leaves demonstrated that nitrate was reduced to ammonia and assimilated into glutamine, glutamate, aspartate and asparagine in the light and in darkness. In the present review, we examined that the reductive incorporation of nitrate into amino acids occurs in darkness in the leaves through the isoforms of NR, NiR, GS, Fd-GOGAT, AspAT and AS. To provide reducing equivalents to the NiR and Fd-GOGAT reactions in the dark, singe leaf contains the photosynthetic form of FNR and Fd and heterotrophic form of FNR and Fd, indicating inter-connected electron supply systems in the light and in darkness. It remains to dissect the operation mechanism of electron donation systems in distinct types of photosynthetic cells and heterotrophic cells of a leaf.”

  1. Just a suggestion, authors can include a section representing effect of abiotic stress on Nitrogen assimilation in light independent conditions or, Do stress hampered photosynthesis facilitates light independent N-assimilation?

[Response]

Response: We introduced abiotic stress experiments on the light independent nitrogen assimilation, in the text (“4.1 Reductant supply for nitrate reductase and nitrite reductase”). The studies concerned the dark stress treatments of the plants, deprived of light by a continuous or prolonged darkness, and measured the impact on Fd and FNR in leaves, necessary for the Fd-NiR and Fd-GOGAT reactions. Also, the text include environmental stress by the high light irradiance on Fd in leaves, leading to an inhibition of photosynthesis and plant growth, as follow:

“Expression studies determined the level of Fd and FNR in the plants deprived of light by continuous or prolonged darkness. This dark stress declined the photosynthetic FNR subforms (pFNRI and pFNRII) at both mRNA and protein levels at the base section of wheat leaf in the presence of nitrate [87], and leaf-type Fds (Fd I and Fd II) in maize leaves [68], suggesting a less efficient contribution of NADPH-FNR-Fd system necessary for the reactions of Fd-NiR and Fd-GOGAT. Exposure of Arabidopsis to environmental stress such as extended high light (120 h at approximately 500 micromol photons m2 sec1) resulted in a gradual decrease of AtFd2 (At1g60950) in its mRNA (to 10% of the WT level) and protein (to 13%) [88]. Down-regulation or mutation of Fd in Arabidopsis [89] and potato (Solanum tuberosum) [90] caused an inactivated photosynthesis and inhibited plant growth.”

Reviewer 3 Report

This review provides important insights on the nitrate assimilation mechanism into amino acids in darkness. 

1) In the introduction section, the authors suggest the relevance of long-distance signaling to the nitrate assimilation mechanism. However, the description regarding this point in this review is limited. It would be nice if the authors could include more description regarding the spatio-temporal nitrate assimilation mechanism.

2) Personally I would prefer that Fig.1 is presented in color.

3) In Fig.2, I am glad that the authors show which regulatory events occur in the light condition, in darkness, or both.

4) In Fig.1 and Fig.2, what meaning did the dashed arrow lines refer to?

Author Response

Reviewer 3

This review provides important insights on the nitrate assimilation mechanism into amino acids in darkness. 

1) In the introduction section, the authors suggest the relevance of long-distance signaling to the nitrate assimilation mechanism. However, the description regarding this point in this review is limited. It would be nice if the authors could include more description regarding the spatio-temporal nitrate assimilation mechanism.

Response: In the introduction, initially long-distance transport addressed to carbohydrate translocation. In response to the reviewer’s comment, nitrate signaling is evoked as a long-distance signaling molecule controlling the nitrate assimilation and plant growth, as follow (in “3.1 Nitrate reduction to ammonia”):

 “Nitrate is not only an essential nutrient but also a signaling molecule of cellular events in response to its fluctuating availability in both space and time. Nitrate is taken up into the roots by the nitrate transporters on the plasma membrane and regulates lateral root developments [16]. Transported in the xylem, nitrate is distributed within the plant by the nitrate transporters located in the shoots, leaves, flowers and seeds, and triggers such as expression of nitrate-responding genes, leaf development and seed germination [17, 18, 19]. Nitrate reductase in the cytosol (Eq. (1)) and ferredoxin (Fd)-dependent nitrite reductase in the chloroplasts/plastids (Eq. (2)) catalyze the sequential reactions of nitrate reduction to nitrite and nitrite to ammonia, respectively (Figure 1).”

2) Personally I would prefer that Fig.1 is presented in color.

Response: Figure 1 is presented in color by showing the chloroplast compartment in green and the electron flow in red. This color presentation is also applied to Figure 2.

3) In Fig.2, I am glad that the authors show which regulatory events occur in the light condition, in darkness, or both.

Response: Figure 2 shows a schematic presentation of electron donation in the light or in darkness for the nitrite reduction and glutamate formation. The electron donation takes place in a population of photosynthetic cells and that of heterotrophic cells within a leaf, and the NADPH-FNR-Fd reducing systems to NiR and GOGAT in the light or in darkness would be under the fine controls of regulatory signals and elements, which are not presented in this schema.

4) In Fig.1 and Fig.2, what meaning did the dashed arrow lines refer to?

Response: The dashed arrow lines indicate the electron flow.

Round 2

Reviewer 1 Report

Thanks for addressing all of my comments and suggestions.

Great job